# Fast Graph Attention Networks Using Effective Resistance Based Graph Sparsification

## Abstract

The attention mechanism has demonstrated superior performance for inference over nodes in graph neural networks (GNNs), however, they result in a high computational burden during both training and inference. We propose FastGAT, a method to make attention based GNNs lightweight by using spectral sparsification to generate an optimal pruning of the input graph. This results in a per-epoch time that is almost linear in the number of graph nodes as opposed to quadratic. We theoretically prove that spectral sparsification preserves the features computed by the GAT model, thereby justifying our FastGAT algorithm. We experimentally evaluate FastGAT on several large real world graph datasets for node classification tasks under both inductive and transductive settings. FastGAT can dramatically reduce (up to **10x**) the computational time and memory requirements, allowing the usage of attention based GNNs on large graphs.

## 1 Introduction

Graphs are efficient representations of pairwise relations, with many real-world applications including product co-purchasing network ((McAuley et al., 2015)), co-author network ((Hamilton et al., 2017b)), etc. Graph neural networks (GNN) have become popular as a tool for inference from graph based data. By leveraging the geometric structure of the graph, GNNs learn improved representations of the graph nodes and edges that can lead to better performance in various inference tasks ((Kipf & Welling, 2016; Hamilton et al., 2017a; Veličković et al., 2018)). More recently, the attention mechanism has demonstrated superior performance for inference over nodes in GNNs ((Veličković et al., 2018; Xinyi & Chen, 2019; Thekumparampil et al., 2018; Lee et al., 2020; Bianchi et al., 2019; Knyazev et al., 2019)). However, attention based GNNs suffer from huge computational cost. This may hinder the applicability of the attention mechanism to large graphs.

GNNs generally rely on graph convolution operations. For a graph $G$ with $N$ nodes, graph convolution with a kernel $\boldsymbol{g_w} : \mathbb{R} \to \mathbb{R}$ is defined as

$$\boldsymbol{g_w} \star \boldsymbol{h} = \boldsymbol{U} \boldsymbol{g_w}(\boldsymbol{\Lambda}) \boldsymbol{U}^\top \boldsymbol{h} \tag{1}$$

where $\boldsymbol{U}$ is the matrix of eigenvectors and $\boldsymbol{\Lambda}$ is the diagonal matrix of the eigenvalues of the normalized graph Laplacian matrix defined as

$$\boldsymbol{L}_{\mathbf{norm}} = \boldsymbol{I} - \boldsymbol{D}^{-1/2} \boldsymbol{A} \boldsymbol{D}^{-1/2}, \tag{2}$$

with $\boldsymbol{D}$ and $\boldsymbol{A}$ being the degree matrix and the adjacency matrix of the graph, and $\boldsymbol{g_w}$ is applied elementwise. Since computing $\boldsymbol{U}$ and $\boldsymbol{\Lambda}$ can be very expensive ($O(N^3)$), most GNNs use an approximation of the graph convolution operator. For example, in graph convolution networks (GCN) (Kipf & Welling, 2016), node features are updated by computing averages as a first order approximation of Eq.equation 1 over the neighbors of the nodes. A single neural network layer is defined as:

$$\boldsymbol{H}_{\mathrm{GCN}}^{(l+1)} = \sigma \left( \widetilde{\boldsymbol{D}}^{-1/2} \widetilde{\boldsymbol{A}} \widetilde{\boldsymbol{D}}^{-1/2} \boldsymbol{H}^{(l)} \boldsymbol{W}^{(l)} \right), \tag{3}$$

where $\boldsymbol{H}^{(l)}$ and $\boldsymbol{W}^{(l)}$ are the activations and the weight matrix at the $l$th layer respectively and $\widetilde{\boldsymbol{A}} = \boldsymbol{A} + \boldsymbol{I}$ and $\widetilde{\boldsymbol{D}}$ is the degree matrix of $\widetilde{\boldsymbol{A}}$.

Attention based GNNs add another layer of complexity: they compute pairwise attention coefficients between all connected nodes. This process can significantly increase the computational burden,

especially on large graphs. Approaches to speed up GNNs were proposed in (Chen et al., 2018; Hamilton et al., 2017a). However, these sampling and aggregation based methods were designed for simple GCNs and are not applicable to attention based GNNs. There has also been works in inducing sparsity in attention based GNNs (Ye & Ji, 2019; Zheng et al., 2020), but they focus on addressing potential overfitting of attention based models rather than scalability.

In this paper, we propose *Fast Graph Attention neTwork* (FastGAT), an edge-sampling based method that leverages effective resistances of edges to make attention based GNNs lightweight. The effective resistance measures importance of the edges in terms of preserving the graph connectivity. FastGAT uses this measure to prune the input graph and generate a randomized subgraph with far fewer edges. Such a procedure preserves the spectral features of a graph, hence retaining the information that the attention based GNNs need. At the same time, the graph is amenable to more complex but computationally intensive models such as attention GNNs. With the sampled subgraph as their inputs, the attention based GNNs enjoy much smaller computational complexity. Note that FastGAT is applicable to all attention based GNNs. In this paper, we mostly focus on the Graph Attention NeTwork model (GAT) proposed by (Veličković et al., 2018). However we also show FastGAT is generalizable to two other attention based GNNs, namely the cosine similarity based approach (Thekumparampil et al., 2018) and Gated Attention Networks (Zhang et al., 2018).

We note that Graph Attention Networks can be re-interpreted as convolution based GNNs. We show this explicitly in the Appendix. Based on this re-interpretation, we theoretically prove that spectral sparsification preserves the feature representations computed by the GAT model. We believe this interpretation also opens up interesting connections between sparsifying state transition matrices of random walks and speeding up computations in GNNs.

The contributions of our paper are as outlined below:

- We propose FastGAT, a method that uses effective resistance based spectral graph sparsification to accelerate attention GNNs in both inductive and transductive learning tasks. The rapid subsampling and the spectrum preserving property of FastGAT help attention GNNs retain their accuracy advantages and become computationally light.

- We provide a theoretical justification for using spectral sparsification in the context of attention based GNNs by proving that spectral sparsification preserves the features computed by GNNs.

- FastGAT outperforms state-of-the-art algorithms across a variety of datasets under both transductive and inductive settings in terms of computation, achieving a speedup of up to **10x** in training and inference time. On larger datasets such as Reddit, the standard GAT model runs out of memory, whereas FastGAT achieves an F1 score 0.93 with 7.73 second per epoch time in training.

- Further, FastGAT is generalizable to other attention based GNNs such as the cosine similarity based attention (Thekumparampil et al., 2018) and the Gated Attention Network (Zhang et al., 2018).

## 2 RELATED WORK

**Accelerating graph based inference** has drawn increasing interest. Two methods proposed in (Chen et al., 2018) (FastGCN) and (Huang et al., 2018) speed up GCNs by using importance sampling to sample a subset of nodes per layer during training. Similarly, GraphSAGE (Hamilton et al., 2017a) also proposes an edge sampling and aggregation based method for inductive learning based tasks. All of the above works use simple aggregation and target simple GCNs, while our work focus on more recent attention based GNNs such as (Veličković et al., 2018). We are able to take advantage of the attention mechanism, while still being computationally efficient.

**Graph sparsification** aims to approximate a given graph by a graph with fewer edges for efficient computation. Depending on final goals, there are cut-sparsifiers ((Benczúr & Karger, 1996)), pairwise distance preserving sparsifiers ((Althöfer et al., 1993)) and spectral sparsifiers ((Spielman & Teng, 2004; Spielman & Srivastava, 2011)) , among others ((Zhao, 2015; Calandriello et al., 2018; Hübler et al., 2008; Eden et al., 2018; Sadhanala et al., 2016)). In this work, we use spectral sparsification to choose a randomized subgraph while preserving spectral properties. Apart form providing the strongest guarantees in preserving graph structure ((Chu et al., 2018)), they align well with GNNs due to their connection to spectral graph convolutions.

**Graph sparsification on neural networks** have been studied recently ((Ye & Ji, 2019; Zheng et al., 2020; Ioannidis et al., 2020; Louizos et al., 2017)). However, their main goal is to alleviate overfitting in GNNs not reducing the training time. They still require learning attention coefficients and binary gate values for all edges in the graph, hence not leading to any computational or memory benefit. In contrast, FastGAT uses a fast subsampling procedure, thus resulting in a drastic improvement in training and inference time. It is also highly stable in terms of training and inference.

## 3 FASTGAT: ACCELERATING GRAPH ATTENTION NETWORKS VIA EDGE SAMPLING

### 3.1 THE FASTGAT ALGORITHM

Let $\boldsymbol{G}(\mathcal{E}, \mathcal{V})$ be a graph with $N$ nodes and $M$ edges. An attention based GNN computes attention coefficients $\alpha_{i,j}$ for every pair of connected nodes $i, j \in \mathcal{V}$ in every layer $\ell$. The $\alpha_{i,j}$'s are then used as averaging weights to compute the layer-wise feature updates. In the original GAT formulation, the attention coefficients are

$$\alpha_{ij} = \frac{\exp\left(\text{LeakyReLU}(\boldsymbol{a}^\top [\boldsymbol{W}\boldsymbol{h_i} || \boldsymbol{W}\boldsymbol{h_j}])\right)}{\sum_{j \in \mathcal{N}_i} \exp\left(\text{LeakyReLU}(\boldsymbol{a}^\top [\boldsymbol{W}\boldsymbol{h_i} || \boldsymbol{W}\boldsymbol{h_j}])\right)}, \tag{4}$$

where $\boldsymbol{h_i}$'s are the input node features to the layer, $\boldsymbol{W}$ and $\boldsymbol{a}$ are linear mappings that are learnt, $\mathcal{N}_i$ denotes the set of neighbors of node $i$, and $||$ denotes concatenation. With the $\alpha_{ij}$'s as defined above, the node-$i$ output embedding of a GAT layer is

$$\boldsymbol{h_i'} = \sigma\left(\sum_{j \in \mathcal{N}_i} \alpha_{ij} \boldsymbol{W}\boldsymbol{h_j}\right). \tag{5}$$

For multi-head attention, the coefficients are computed independently in each attention head with head-dependent matrices $\boldsymbol{W}$ and attention vector $\boldsymbol{a}$. Note that the computational burden in GATs arises directly from computing the $\alpha_{i,j}$'s in *every layer, every attention head and every forward pass* during training.

**Goal:** Our objective is to achieve performance equivalent to that of full graph attention networks (GAT), but with only a fraction of the original computational complexity. This computational saving is achieved by reducing the number of attention computations.

**Idea:** We propose to use **edge-sampling functions** that sparsify graphs by removing nonessential edges. This leads to direct reduction in the number of attention coefficients to be computed, hence reducing the burden. Choosing the sampling function is crucial for retaining the graph connectivity.

Let $\text{EdgeSample}(E, \boldsymbol{A}, q)$ denote a *randomized sampling function* that, given an edge set $E$, adjacency matrix $\boldsymbol{A}$ and a number of edges to be sampled $q$, returns a subset of the original edge set $E_s \subset E$ with $|E_s| = q$. Our algorithm then uses this function to sparsify the graph in every layer and attention head. Following this, the attention coefficients are computed *only for the remaining edges*. A more detailed description is given in Algorithm 1. In every layer and attention head, a randomized subgraph with $q \ll M$ edges is generated and the attention coeffients are computed only for this subset of edges. We use a specialized distribution that depends on the contribution of each edge to the graph connectivity. We provide further details in Section 3.2.

Note that in the general description below, the attention coefficients themselves are used as weights for sparsification and the reweighted attention coefficients are used to compute the feature update. Doing so helps in theoretical analysis of the algorithm. However in practice, we replace this expensive procedure with a one-time sampling of the graph with the original edge weights and compute the attention coefficients for only the remaining edges. In particular, we use two simpler variations of FastGAT include: i) FastGAT-const, where the subgraph $g$ is kept constant in all the layers and attention heads and ii) FastGAT-layer, where the subgraph is different in each layer (drawn stochastically from the original edge weights), but the same across all the attention heads within a layer.

---

**Algorithm 1:** The FastGAT Algorithm

---

**Input:** Graph $G(\mathcal{V}, \mathcal{E})$, Num. layers = $L$, Num. Attention heads $K^{(\ell)}$, $\ell = 1, \cdots, L$
    Initial Weight matrices $\boldsymbol{W}^{(\ell)}$, Non-linearity $\sigma$, Feature matrix $H \in \mathbb{R}^{N \times D}$
    Randomized edge sampling function EdgeSample$(\cdot)$, Attention function $\phi(\cdot)$
    Num. edges sampled $q$

**for** *each layer $\ell$* **do**

    **for** *each attention head $k \in \{1, 2, \cdots, K^{(\ell)}\}$* **do**

        Compute attention matrix $\Gamma_k^{(\ell)} \in \mathbb{R}^{N \times N}$, with $\Gamma_k^{(\ell)}(i, j) = \phi_{\theta_k}(\boldsymbol{h}_i^{(\ell)}, \boldsymbol{h}_j^{(\ell)})$

        Sample a graph $\hat{\Gamma}_k^{(\ell)} = \text{EdgeSample}(\Gamma_k^{(\ell)}, \boldsymbol{A}, q)$

        Compute $\boldsymbol{H}_k^{(\ell+1)} = \sigma\left(\hat{\Gamma}_k^{(\ell)} \boldsymbol{H}_k^{(\ell)} \boldsymbol{W}^{(\ell)}\right)$

    $\boldsymbol{H}^{(\ell+1)} = \underset{k}{||} \boldsymbol{H}_k^{(\ell)}$ // Concatenate the output of attention heads

Compute loss and update $\boldsymbol{W}$'s // gradient based weight update

---

## 3.2 SAMPLING GRAPH EDGES USING EFFECTIVE RESISTANCES

We use a particular edge sampling function EdgeSample$(\cdot)$ that is motivated by the field of spectral graph sparsification. Let $\boldsymbol{L}$ represent the graph Laplacian (defined as $\boldsymbol{L} = \boldsymbol{D} - \boldsymbol{A}$ where $\boldsymbol{D}$ is the degree matrix), $\lambda_i(\boldsymbol{L})$ denote the $i$th eigenvalue of $\boldsymbol{L}$ and let $\boldsymbol{A}^\dagger$ denote the Moore-Penrose inverse of a matrix.

Motivated by the fact that GNNs are approximations of spectral graph convolutions (defined in equation 1), we aim to preserve the spectrum (or eigenstructure) of the graph. Formally, let $\boldsymbol{L_G}$ and $\boldsymbol{L_H}$ be the Laplacian matrices of the original graph $\boldsymbol{G}$ and the sparsified graph $\boldsymbol{H}$. Then, spectral graph sparsification ensures that the spectral content of $\boldsymbol{H}$ is similar to that of $\boldsymbol{G}$:

$$(1 - \epsilon)\lambda_i(\boldsymbol{L_G}) \leq \lambda_i(\boldsymbol{L_H}) \leq (1 + \epsilon)\lambda_i(\boldsymbol{L_G}), \ \forall i \tag{6}$$

where $\epsilon$ is any desired threshold. (Spielman & Srivastava, 2011) showed how to achieve this by using a distribution proportional to the effective resistances of the edges

**Definition 1 (Effective Resistance)** *(Spielman & Srivastava, 2011) The effective resistance between any two nodes of a graph can be defined as the potential difference induced across the two nodes, when a unit current is induced at one node and extracted from the other node. Mathematically, it is defined as below.*

$$R_e(u, v) = \boldsymbol{b}_e^\top \boldsymbol{L}^\dagger \boldsymbol{b}_e,$$

*where $\boldsymbol{b_e} = \boldsymbol{\chi_u} - \boldsymbol{\chi_v}$ ($\chi_l$ is a standard basis vector with $1$ in the lth position) and $\boldsymbol{L}^\dagger$ is the pseudo-inverse of the graph Laplacian matrix.*

The effective resistance measures the importance of an edge to the graph structure. For example, the removal of an edge with high effective resistance can harm the graph connectivity. The particular function EdgeSample we use in FastGAT is described in Algorithm 2.

---

**Algorithm 2:** Effective resistance based EdgeSample function (Spielman & Srivastava, 2011)

---

**Input**: Graph $\boldsymbol{G}(\mathcal{E}_G, \mathcal{V})$, $w_e$ is the edge weight for $e \in \mathcal{E}$, $0 < \epsilon < 1$

For each edge $e(u, v)$, compute $R_e(u, v)$ using fast algorithm in (Spielman & Srivastava, 2011)

Set $q = \max(M, \text{int}(0.16N \log N/\epsilon^2))$, $\boldsymbol{H} = \text{Graph}(\mathcal{E}_H = \text{Null}, \mathcal{V})$

**for** $i \leftarrow 1$ **to** $q$ **do**

    Sample an edge $e_i$ from the distribution $p_e$ proportional to $w_e R_e$

    **if** $e_i \in \mathcal{E}_H$ **then**

        Add $w_e/qp_e$ to its weight    // Increase weight of an existing edge

    **else** Add $e_i$ to $\mathcal{E}_H$ with weight $w_e/qp_e$  // Add the edge for the first time

$\boldsymbol{H} = \text{Graph}(\mathcal{E}_H, \mathcal{V})$

---

The effective-resistance based edge-samplng function is described in Algorithm 2. For a desired value of $\epsilon$, the algorithm sampled $q = O(N \log N/\epsilon^2)$ number of edges such that equation 6 is satisfied.

**Choosing** $\epsilon$. As shown in Algorithm. 2, it requires setting a pruning parameter $\epsilon$, which determines the quality of approximation after sparsification and also determines the number of edges retained $q$. The choice of $\epsilon$ is a design parameter at the discretion of the user. To remove the burden of choosing $\epsilon$, we also provide an **adaptive algorithm** in Section B.4 in the appendix.

**Complexity**. The sample complexity $q$ in Algorithm. 2 directly determines the final complexity. If $q = O(N \log N/\epsilon^2)$, then the spectral approximation in equation 6 can be achieved (Spielman & Srivastava, 2011). Note that this results in a number of edges that is almost linear in the number of nodes, as compared to quadratic as in the case of dense graphs. The complexity of computing $R_e$ for all edges (up to a constant factor approximation) is $O(M \log N)$ time, where $M$ is the number of edges (Spielman & Srivastava, 2011). While we describe the algorithm in detail in the appendix (Section B.3) , it uses a combination of fast solutions to Laplacian based linear systems and the Johnson-Lindenstrauss Lemma [1]. This is almost linear in the number of edges, and hence much smaller than the complexity of computing attention coefficients in every layer and forward pass of GNNs. Another important point is that the computation of $R_e$'s is a one-time cost. Unlike graph attention coefficients, we do not need to recompute the effective resistances in every training iteration. Hence, once sparsified, the same graph can be used in all subsequent experiments. Further, since each edge is sampled independently, the edge sampling process itself can be parallelized.

## 4    THEORETICAL ANALYSIS OF FASTGAT

In this section we provide the theoretical analysis of FastGAT. Although we used the sampling strategy provided in (Spielman & Srivastava, 2011), their work address the preservation of only the eigenvalues of $\boldsymbol{L}$. However, we are interested in the following question: *Can preserving the spectral structure of the graph lead to good performance under the GAT model?* To answer this question, we give an upper bound on the error between the feature updates computed by a single layer of the GAT model using the full graph and a sparsified graph produced by FastGAT.

Spectral sparsification preserves the spectrum of the underlying graph. This then hints that neural network computations that utilize spectral convolutions can be approximated by using sparser graphs. We first show that this is true in a layer-wise sense for the GCN (Kipf & Welling, 2016) model and then show a similar result for the GAT model as well. Below, we use ReLU to denote the standard Rectified Linear Unit and ELU to denote the Exponential Linear Unit.

**Theorem 1** *At any layer $l$ of a GCN model with input features $\boldsymbol{H}^{(l)} \in \mathbb{R}^{N \times D}$, weight matrix $\boldsymbol{W}^{(l)} \in \mathbb{R}^{D \times F}$, if the element-wise non-linearity function $\sigma$ is either the ReLU or the ELU function, the features $\widehat{\boldsymbol{H}_f}$ and $\widehat{\boldsymbol{H}_s}$ computed using equation 3 with the full and a layer dependent spectrally sparsified graph obey*

$$\left\| \widehat{\boldsymbol{H}_f} - \widehat{\boldsymbol{H}_s} \right\|_F \le 8\epsilon \left\| \boldsymbol{H}^{(l)} \boldsymbol{W}^{(l)} \right\|_F. \tag{7}$$

*where $\boldsymbol{L}_{\mathbf{norm}}$ is as defined in equation 2 and $\|\cdot\|$ denotes the spectral norm.*

In our next result, we show a similar upper bound on the features computed with the full and the sparsified graphs using the GAT model.

**Theorem 2** *At any layer $l$ of GAT with input features $\boldsymbol{H}^{(\ell)} \in \mathbb{R}^{N \times D}$, weight matrix $\boldsymbol{W}^{(l)} \in \mathbb{R}^{D \times F}$ and $\alpha_i j$'s be the attention coefficients in that layer. Let the non-linearity used by either ReLU or the ELU functon. Then, the features $\widehat{\boldsymbol{H}_f}$ and $\widehat{\boldsymbol{H}_s}$ computed using equation 5 with the full and a layer dependent spectrally sparsified graph obey*

$$\left\| \widehat{\boldsymbol{H}_f} - \widehat{\boldsymbol{H}_s} \right\|_F \le 12\epsilon \left\| \boldsymbol{H}^{(l)} \boldsymbol{W}^{(l)} \right\|_F \tag{8}$$

*where $\|\cdot\|$ denotes the spectral norm of the matrix.*

---

[1]https://en.wikipedia.org/wiki/Johnson-Lindenstrauss_lemma

Theorem 2 shows that our proposed layer-wise spectral sparsification leads to good approximations of latent embedding $\widehat{H}$ for GAT model as well. The guarantees given above assume a layer-wise sparsification that is updated based on the attention coefficients. To circumvent the associated computational burden, we use the simpler versions such as 1-const and always use the original weight matrix to sparsify the graph in each layer. In the experiment section, we show that such a relaxation by a one-time spectral sparsification does not lead to any degradation in performance.

**Approximation of weight matrices**. Theorems 1 and 2 provide an upper bound on the feature updates obtained using the full and sparsified graphs. In practice, we observe an even stronger notion of approximation between GAT and FastGAT: the weight matrices of the two models post training are good approximations of each other. We report this observation in Section. A.4 in the appendix. We show that the error between the learned matrices is small and proportional to the value of $\epsilon$ itself.

## 5 EXPERIMENTS

**Datasets** We evaluated FastGAT on large graph datasets using semi-supervised node classification tasks. This is a standard task to evaluate GNNs, as done in (Veličković et al., 2018; Hamilton et al., 2017a; Kipf & Welling, 2016). Datasets are sourced from the DGLGraph library (DGL). Their statistics are provided in Table 1. We evaluate on both transductive and inductive tasks. The PPI dataset serves as a standard benchmark for inductive classification and the rest of the datasets for transductive classification. Further details about the datasets including details about train/validaton/ test split are given in the appendix (Section B.1). We also evaluated on smaller datasets including Cora, citeseer and Pubmed, but present their results in the appendix (Section B.2).

Table 1: Dataset Statistics

| Dataset | Reddit | Coauth-Phy | Github | Amaz.Comp | Coauth-cs . | Amaz.Photos | PPI (Inductive task) |
|---|---|---|---|---|---|---|---|
| Nodes | 232,965 | 34,493 | 37,700 | 13,752 | 18,333 | 7,650 | 56944 (24 graphs) |
| Edges | 57 mil | 495,924 | 289,003 | 287,209 | 163778 | 143,662 | 818716 |
| Classes | 41 | 5 | 2 | 10 | 15 | 8 | 121 (multilabel) |

**Baselines**. **Transductive learning:** We compare FastGAT with the following baseline methods. (1) The original graph attention networks (GAT) (Veličković et al., 2018), (2) SparseGAT (Ye & Ji, 2019) that learns edge coefficients to sparsify the graph, (3) random subsampling of edges, and (4) FastGCN (Chen et al., 2018) that is also designed for GNN speedup. Note that we compare SparseGAT in a transductive setting, whereas the original paper (Ye & Ji, 2019) uses an inductive setting. We thus demonstrate that FastGAT can handle the full input graph, unlike any previous attention based baseline method. **Inductive learning:** For this task, we compare with both GAT (Veličković et al., 2018) and GraphSAGE (Hamilton et al., 2017a). More importantly, for both inductive and transductive tasks, we show that a uniform random subsampling of edges results in a drop in performance, where as FastGAT does not.

**Evaluation setup** and **Model Implementation Details** are provided in Section. B in the appendix.

Q1. FASTGAT PROVIDES FASTER TRAINING WITH STATE-OF-THE-ART ACCURACY.

Our first experiment is to study FastGAT on the accuracy and time performance of attention based GNNs in node classification. We sample $q = \text{int}(0.16N \log N/\epsilon^2)$ number of edges from the distribution $p_e$ with replacement, as described in Section 3.2.

**Transductive learning:** In this setting, we assume that the features of all the nodes in the graph, including train, validation and test nodes are available, but only the labels of training nodes are available during training, similar to (Veličković et al., 2018). First, we provide a direct comparison between FastGAT and the original GAT model and report the results in Table 2. As can be observed from the results, FastGAT achieves the same test accuracy as the full GAT across all datasets, while being dramatically faster: we are able to achieve up to **5x** on GPU (**10x on CPU**) speedup.

We then compare FastGAT with the following baselines: SparseGAT (Ye & Ji, 2019), random subsampling of edges and FastGCN (Chen et al., 2018) in Table 3. SparseGAT uses the attention mechanism to learn embeddings and a sparsifying mask on the edges. We compare the training

time per epoch for the baseline methods against FastGAT in Figure 1. The results shows that **FastGAT matches state-of-the-art accuracy (F1-score), while being much faster**. While random subsampling of edges leads to a model that is as fast as ours but with a degradation in accuracy performance. FastGAT is also faster compared to FastGCN on some large datasets, even though FastGCN does not compute any attention coefficients. Overall the classification accuracy of FastGAT remains the same (or sometimes even improves) compared to standard GAT, while the training time reduces drastically. This is most evident in the case of the Reddit dataset, where the vanilla GAT model **runs out of memory** on a machine with 128GB RAM and a Tesla P100 GPU when computing attention coefficients over 57 million edges, while FastGAT can train with 10 seconds per epoch.

Table 2: Comparison of FastGAT with GAT (Veličković et al., 2018).

| Metric | Method | Reddit | Phy | Git | Comp | CS | Photo |
|--------|--------|--------|-----|-----|------|-----|-------|
| F1-micro | GAT | OOM | 0.94±0.001 | 0.86±0.000 | 0.89±0.004 | 0.89±0.001 | 0.92±0.001 |
| | FastGAT-const-0.5 | 0.93±0.000 | 0.94±0.001 | 0.86±0.001 | 0.88±0.004 | 0.88±0.001 | 0.91±0.002 |
| | FastGAT-const-0.9 | 0.88±0.001 | 0.94±0.002 | 0.85±0.002 | 0.86±0.002 | 0.88±0.004 | 0.89±0.002 |
| GPU Time (s) | GAT | OOM | 3.67 | 3.71 | 2.93 | 1.61 | 1.88 |
| | FastGAT-const-0.5 | 7.73 | 1.96 | 2.06 | 0.83 | 1.14 | 0.66 |
| | FastGAT-const-0.9 | 4.07 | 1.80 | 1.63 | 0.50 | 0.95 | 0.40 |
| CPU Time (s) | GAT | OOM | 25.19 | 18.05 | 16.58 | 3.59 | 6.57 |
| | FastGAT-const-0.5 | 178.79 | 4.42 | 5.92 | 2.58 | 2.27 | 1.72 |
| | FastGAT-const-0.9 | 41.42 | 2.70 | 3.44 | 1.22 | 1.48 | 0.77 |
| % Edges redu. | FastGAT-const-0.5 | 97.03% | 74.04% | 55.6% | 78.6% | 67% | 79.3% |
| | FastGAT-const-0.9 | 99.03% | 88.48% | 79.77% | 91.99% | 83.80% | 91.86% |

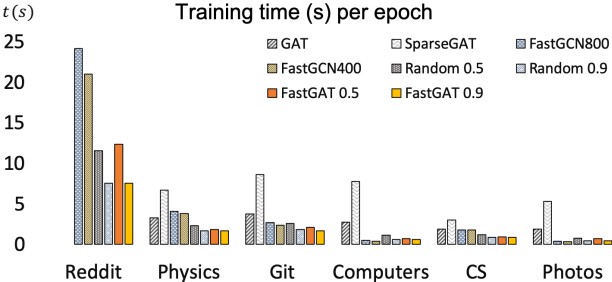

Figure 1: Training time per epoch for our FastGAT model and baselines. FastGAT has a significantly lower training time, while also preserving classification accuracy (in Table. 3). On the Reddit dataset, both GAT and sparseGAT models run out of memory in the transductive setting and hence we do not include bar graphs for them in the plot above.

Table 3: **Transductive learning:** While significantly lower training time (in Fig. 1), FastGAT still has comparable and sometimes better accuracy than other models. Computationally intensive model sparseGAT runs out of memory on the Reddit dataset.

| Method | Reddit | Coauthor-Phy | Github | Amazon-Comp | Coauthor-CS | Amazon-Photos |
|--------|--------|--------------|--------|-------------|-------------|---------------|
| FastGCN-400 (Chen et al., 2018) | 0.77±0.001 | **0.95±0.001** | **0.86±0.001** | 0.81±0.002 | **0.92±0.001** | 0.86±0.005 |
| FastGCN-800 (Chen et al., 2018) | 0.81±0.001 | **0.95±0.001** | **0.86±0.001** | 0.80±0.009 | 0.92±0.001 | 0.86±0.007 |
| GAT rand 0.5 | 0.87±0.002 | 0.94±0.001 | 0.85±0.001 | 0.86±0.003 | 0.88±0.002 | 0.90±0.005 |
| GAT rand 0.9 | 0.82±0.005 | 0.93±0.000 | 0.84±0.002 | 0.83±0.004 | 0.88±0.002 | 0.89±0.004 |
| sparseGAT (Ye & Ji, 2019) | OOM | 0.94±0.004 | 0.84±0.002 | 0.85±0.001 | 0.88±0.002 | **0.91±0.003** |
| FastGAT-layer-0.5 | **0.93±0.001** | **0.95±0.001** | **0.86±0.002** | **0.88 ±0.003** | 0.88±0.002 | 0.90±0.002 |
| FastGAT-layer-0.9 | 0.91±0.002 | 0.94±0.002 | **0.86±0.000** | 0.85±0.002 | 0.88±0.002 | 0.90±0.002 |
| FastGAT-const-0.5 | **0.93±0.000** | 0.94±0.001 | **0.86±0.001** | **0.88±0.004** | 0.88±0.001 | **0.91±0.002** |
| FastGAT-const-0.9 | 0.88±0.001 | 0.94±0.002 | 0.85±0.002 | 0.86±0.002 | 0.88±0.004 | 0.89±0.002 |

**Inductive learning**. FastGAT can also be applied to the inductive learning framework, where features of only the training nodes are available and training is performed using the subgraph consisting of

In Tables 2 and 3, FastGCN-400 denotes that we sample 400 nodes in every forward pass, as described in (Chen et al., 2018) (similarly, in FastGCN-800, we sample 800 nodes). FastGAT-0.5 denotes we use $\epsilon = 0.5$. GAT rand 0.5 uses random subsampling of edges, but keeps the same number of edges as FastGAT-0.5

the training nodes. To show the utility of FastGAT in such a setting, we use the Protein-Protein interaction (PPI) dataset ((Zitnik & Leskovec, 2017)). Our model parameters are the same as in (Veličković et al., 2018), but we sparsify each of the 20 training graphs before training on them. The other 4 graphs are used for validation and testing (2 each). We use $\epsilon = 0.25, 0.5$ in our experiments, since the PPI dataset is smaller than the other datasets. We report the experimental results in Table 4. FastGAT clearly outperforms the baselines like GraphSAGE and uniform subsampling of edges. While it has the same accuracy performance as the GAT model (which is expected), it has a much smaler training time, as reported in Table 4.

Table 4: **Inductive learning on PPI Dataset** The training time per epoch for the full GAT method is around 45.8s, whereas FastGAT requires only about 29.8s when $\epsilon = 0.25$ and 20s when $\epsilon = 0.5$. FastGAT achieved the same accuracy as GAT, and outperforms the other baselines, while also begin computatioanally efficient. The F1 score for GraphSAGE was obtained from what is reported in Veličković et al. (2018).

| Method | F1 score |
|---|---|
| GAT | **0.974 ±0.002** |
| GraphSAGE* | 0.768 ±0.000 |
| GAT rand-0.25 | 0.95 ±0.005 |
| GAT rand-0.9 | 0.7±0.004 |
| FastGAT-const-0.25 | **0.974 ± 0.003** |
| FastGAT-const-0.5 | 0.800 ± 0.005 |

Q2. FASTGAT CAN BE APPLIED TO OTHER ATTENTION BASED GRAPH INFERENCE METHODS.

Finally, we study if FastGAT is sensitive to the particular formulation of the attention function. There have been alternative formulations proposed to capture pairwise similarity. For example, (Thekumparampil et al., 2018) proposes a **cosine similarity** based approach, where the attention coefficient of an edge is defined in equation 9,

$$\alpha_{ij}^{(\ell)} = \underset{j \in \mathcal{N}_i}{\mathrm{softmax}} \left( [\beta^{(\ell)} \cos(\boldsymbol{h}_i^{(\ell)}, \boldsymbol{h}_j^{(\ell)})] \right) \tag{9}$$

where $\beta^{(\ell)}$ is a layer-wise learnable parameter and $\cos(\boldsymbol{x}, \boldsymbol{y}) = \boldsymbol{x}^\top \boldsymbol{y} / \|\boldsymbol{x}\| \|\boldsymbol{y}\|$. Another definition is proposed in (Zhang et al., 2018) (**GaAN**:Gated Attention Networks), which defines attention as in equation 10,

$$\alpha_{ij}^{(l)} = \underset{j \in \mathcal{N}_i}{\mathrm{softmax}} \langle \mathrm{FC}_{src}(\boldsymbol{h}_i^\ell), \mathrm{FC}_{dst}(\boldsymbol{h}_j^\ell) \rangle \tag{10}$$

where $\mathrm{FC}_{src}$ and $\mathrm{FC}_{dst}$ are 2-layered fully connected neural networks.

We performed similar experiments on these attention definitions. Tables. 5 confirmed that FastGAT generalizes to different attention functions.Note that the variability in accuracy performance across Tables 2 and 5 comes from the different definitions of the attention functions and not from FastGAT. Our goal is to show that given a specific GAT model, FastGAT can achieve similar accuracy performance as that model, but in much faster time.

## 6 CONCLUSION

In this paper, we introduced FastGAT, a method to make attention based GNNs lightweight by using spectral sparsification. We theoretically justified our FastGAT algorithm. FastGAT can significantly reduce the computational time across multiple large real world graph datasets while attaining state-of-the-art performance.

## REFERENCES

Deep graph library. `https://docs.dgl.ai/en/0.4.x/index.html`.

Table 5: Comparison of full and sparsified graphs.

| FastGAT for Cosine Similarity | | Reddit | Phy | Github | Comp | CS | Photo |
|---|---|---|---|---|---|---|---|
| F1-micro | Full graph | OOM | 0.96±0.001 | 0.86±0.001 | 0.88±0.003 | 0.91±0.001 | 0.93±0.001 |
| | $\epsilon = 0.5$ | 0.93±0.000 | 0.95±0.001 | 0.86±0.001 | 0.88±0.003 | 0.91±0.001 | 0.92±0.002 |
| | $\epsilon = 0.9$ | 0.88±0.001 | 0.95±0.003 | 0.86±0.002 | 0.87±0.004 | 0.89±0.002 | 0.90±0.002 |
| Time/epoch (s) | Full graph | OOM | 3.39 | 3.89 | 2.80 | 1.54 | 1.89 |
| | $\epsilon = 0.5$ | 8.02 | 1.99 | 2.3 | 0.77 | 1.11 | 0.724 |
| | $\epsilon = 0.9$ | 4.46 | 1.79 | 1.69 | 0.48 | 0.91 | 0.435 |
| **FastGAT for GaAN** | | Reddit | Phy | Github | Comp | CS | Photo |
| F1-micro | Full graph | OOM | 0.94±0.001 | 0.86±0001 | 0.86±0.002 | 0.873±0.001 | 0.90±0.001 |
| | $\epsilon = 0.5$ | 0.92±0.001 | 0.94±0.002 | 0.86±0.002 | 0.84±0.001 | 0.87±0.002 | 0.89±0.003 |
| | $\epsilon = 0.9$ | 0.87±0.001 | 0.93±0.001 | 0.86±0.002 | 0.82±0.002 | 0.84 ±0.001 | 0.88±0.002 |
| Time/epoch (s) | Full graph | OOM | 3.55 | 3.87 | 2.77 | 1.60 | 1.90 |
| | $\epsilon = 0.5$ | 8.10 | 1.93 | 2.10 | 0.84 | 1.10 | 0.67 |
| | $\epsilon = 0.9$ | 4.47 | 1.76 | 1.54 | 0.50 | 0.90 | 0.41 |

Laplacians julia library. https://danspielman.github.io/Laplacians.jl/v0.1/.

Ingo Althöfer, Gautam Das, David Dobkin, Deborah Joseph, and José Soares. On sparse spanners of weighted graphs. *Discrete & Computational Geometry*, 9(1):81–100, 1993.

András A. Benczúr and David R. Karger. Approximating s-t minimum cuts in Õ(n2) time. In *Proceedings of the Twenty-Eighth Annual ACM Symposium on Theory of Computing*, STOC '96, pp. 47–55, New York, NY, USA, 1996. Association for Computing Machinery. ISBN 0897917855. doi: 10.1145/237814.237827. URL https://doi.org/10.1145/237814.237827.

Filippo Maria Bianchi, Daniele Grattarola, Lorenzo Livi, and Cesare Alippi. Hierarchical representation learning in graph neural networks with node decimation pooling. *arXiv preprint arXiv:1910.11436*, 2019.

Daniele Calandriello, Ioannis Koutis, Alessandro Lazaric, and Michal Valko. Improved large-scale graph learning through ridge spectral sparsification. 2018.

Jie Chen, Tengfei Ma, and Cao Xiao. Fastgcn: fast learning with graph convolutional networks via importance sampling. *International Conference on Learning Representations (ICLR), 2018, Vancouver, Canada*, 2018.

Timothy Chu, Yu Gao, Richard Peng, Sushant Sachdeva, Saurabh Sawlani, and Junxing Wang. Graph sparsification, spectral sketches, and faster resistance computation, via short cycle decompositions. In *2018 IEEE 59th Annual Symposium on Foundations of Computer Science (FOCS)*, pp. 361–372. IEEE, 2018.

Talya Eden, Shweta Jain, Ali Pinar, Dana Ron, and C Seshadhri. Provable and practical approximations for the degree distribution using sublinear graph samples. In *Proceedings of the 2018 World Wide Web Conference*, pp. 449–458, 2018.

Will Hamilton, Zhitao Ying, and Jure Leskovec. Inductive representation learning on large graphs. In *Advances in neural information processing systems*, pp. 1024–1034, 2017a.

William L. Hamilton, Rex Ying, and Jure Leskovec. Inductive representation learning on large graphs. *CoRR*, abs/1706.02216, 2017b.

Wenbing Huang, Tong Zhang, Yu Rong, and Junzhou Huang. Adaptive sampling towards fast graph representation learning. In *Advances in neural information processing systems*, pp. 4558–4567, 2018.

Christian Hübler, Hans-Peter Kriegel, Karsten Borgwardt, and Zoubin Ghahramani. Metropolis algorithms for representative subgraph sampling. In *2008 Eighth IEEE International Conference on Data Mining*, pp. 283–292. IEEE, 2008.

Vassilis N. Ioannidis, Siheng Chen, and Georgios B. Giannakis. Pruned graph scattering transforms. In *International Conference on Learning Representations*, 2020. URL https://openreview.net/forum?id=rJeg7TEYwB.

Thomas N Kipf and Max Welling. Semi-supervised classification with graph convolutional networks. *arXiv preprint arXiv:1609.02907*, 2016.

Boris Knyazev, Graham W Taylor, and Mohamed Amer. Understanding attention and generalization in graph neural networks. In *Advances in Neural Information Processing Systems*, pp. 4204–4214, 2019.

Bohyun Lee, Shuo Zhang, Aleksandar Poleksic, and Lei Xie. Heterogeneous multi-layered network model for omics data integration and analysis. *Frontiers in Genetics*, 10:1381, 2020.

Qimai Li, Zhichao Han, and Xiao-Ming Wu. Deeper insights into graph convolutional networks for semi-supervised learning. In *Thirty-Second AAAI Conference on Artificial Intelligence*, 2018.

Christos Louizos, Max Welling, and Diederik P Kingma. Learning sparse neural networks through $l\_0$ regularization. *arXiv preprint arXiv:1712.01312*, 2017.

Julian McAuley, Rahul Pandey, and Jure Leskovec. Inferring networks of substitutable and complementary products. In *Proceedings of the 21th ACM SIGKDD International Conference on Knowledge Discovery and Data Mining*, KDD '15, pp. 785–794, New York, NY, USA, 2015. Association for Computing Machinery. ISBN 9781450336642. doi: 10.1145/2783258.2783381. URL https://doi.org/10.1145/2783258.2783381.

Veeru Sadhanala, Yu-Xiang Wang, and Ryan Tibshirani. Graph sparsification approaches for laplacian smoothing. In Arthur Gretton and Christian C. Robert (eds.), *Proceedings of the 19th International Conference on Artificial Intelligence and Statistics*, volume 51 of *Proceedings of Machine Learning Research*, pp. 1250–1259, Cadiz, Spain, 09–11 May 2016. PMLR. URL http://proceedings.mlr.press/v51/sadhanala16.html.

Daniel A. Spielman and Nikhil. Srivastava. Graph sparsification by effective resistances. *SIAM Journal on Computing*, 40(6):1913–1926, 2011. doi: 10.1137/080734029. URL https://doi.org/10.1137/080734029.

Daniel A Spielman and Shang-Hua Teng. Nearly-linear time algorithms for graph partitioning, graph sparsification, and solving linear systems. In *Proceedings of the thirty-sixth annual ACM symposium on Theory of computing*, pp. 81–90, 2004.

Daniel A Spielman and Shang-Hua Teng. Spectral sparsification of graphs. *SIAM Journal on Computing*, 40(4):981–1025, 2011.

Gabriel Taubin. A signal processing approach to fair surface design. In *Proceedings of the 22nd annual conference on Computer graphics and interactive techniques*, pp. 351–358, 1995.

Kiran K Thekumparampil, Chong Wang, Sewoong Oh, and Li-Jia Li. Attention-based graph neural network for semi-supervised learning. *arXiv preprint arXiv:1803.03735*, 2018.

Petar Veličković, Guillem Cucurull, Arantxa Casanova, Adriana Romero, Pietro Liò, and Yoshua Bengio. Graph Attention Networks. *International Conference on Learning Representations*, 2018. URL https://openreview.net/forum?id=rJXMpikCZ.

Zhang Xinyi and Lihui Chen. Capsule graph neural network. In *International Conference on Learning Representations*, 2019. URL https://openreview.net/forum?id=Byl8BnRcYm.

Yang Ye and Shihao Ji. Sparse graph attention networks. *arXiv preprint arXiv:1912.00552*, 2019.

Jiani Zhang, Xingjian Shi, Junyuan Xie, Hao Ma, Irwin King, and Dit-Yan Yeung. Gaan: Gated attention networks for learning on large and spatiotemporal graphs. *arXiv preprint arXiv:1803.07294*, 2018.

Peixiang Zhao. gsparsify: Graph motif based sparsification for graph clustering. In *Proceedings of the 24th ACM International on Conference on Information and Knowledge Management*, pp. 373–382, 2015.

Cheng Zheng, Bo Zong, Wei Cheng, Dongjin Song, Jingchao Ni, Wenchao Yu, Haifeng Chen, and Wei Wang. Robust graph representation learning via neural sparsification, 2020. URL https://openreview.net/forum?id=S1emOTNKvS.

Marinka Zitnik and Jure Leskovec. Predicting multicellular function through multi-layer tissue networks. *Bioinformatics*, 33(14):i190–i198, 2017.

