# OpenReview forum: "FAST GRAPH ATTENTION NETWORKS USING EFFECTIVE RESISTANCE BASED GRAPH SPARSIFICATION"
_ICLR.cc/2021/Conference — Reject_

### Official Review · AnonReviewer2 · 2020-10-24
**Generally reasonable approach, some questions**

**Rating:** 6
**Confidence:** 3

**Review:**

Summary: This paper suggests speeding up attention in graph neural networks by computing a spectral sparsifier of the graph, and letting each node attend only to its neighbors in the sparsifier. Some theoretical justifications are presented. The experimental results show that the accuracy remains about the same while gaining computational efficiency.

Comments:
Section 3 - Clarification questions on your algorithm:
1. Do you denote by A the weighted or unweighted adjacency matrix?
2. Do you sparsify the original graph as a weighted or unweighted graph (and if weighted then with what weights)?
3. In section 3.1 you describe your algorithm as using an edge sampling procedure that returns a subset E_s of the input edges E. However. spectral sparsification returns a weighted graph (regardless of whether the original graph was weighted or unweighted), so in addition to E_s you get new edge weights. Where are those sparsifier weights used in the FastGAT algorithm (algorithm 1)? If they aren't used then the theoretical results would not hold.

Section 4 - The novelty of the theoretical results is perhaps somewhat oversold, as they follow rather immediately from the definition of spectral sparsification. It is not quite true that Spielman-Srivastava "address the preservation of only theeigenvalues of L", as their result preserves the Laplacian quadratic form for any vector (which is indeed what you use in the proof of theorems 1 and 2). Small remarks on this section:
1. The statements of theorems 1,2 are inconsistent in whether to use the superscript "(l)" for H and W.
2. "The complexity of computing R_e for all edges is O(M log N) time" - such result is not known, what Spielman-Srivastava compute in that time complexity is a constant approximation of each R_e, which is sufficient for their importance-sampling spectral sparsification scheme as long as edges are oversampled accordingly.
3. By the way, ||L_norm|| (spectral norm) is at most 2 for any graph.

Overall: The approach generally makes sense and the experiments show benefit, so pending some clarifications about the algorithm requested above, I think the paper can be accepted.

=== Post-discussion update ===

I thank the authors for engaging in the discussion.

I am left somewhat divided on the paper. During the discussion phase, the statement of the main algorithm in the paper changed quite significantly: in the original version each node could attend only to its neighbors in the sparsifier, while in the current version each node attends to all its neighbors (i.e., full graph attention is computed), and the sparsifier is only used in the subsequent feature update. The authors eventually explained that their analysis holds for the latter algorithm (even though its running time is not faster than non-sparsified GAT), while the former algorithm is what they actually implement since it has better running time (albeit no formal guarantees).

I don't take issue with the divide between the theory and the implementation (as long as it is made clear in the paper). I do think, however, that perhaps the theoretical content ended up doing the paper more harm than good. Effective resistances measure the "importance" of edges to the connectivity of the graph - this is a general phenomenon, and the sparsification algorithm of Spielman-Srivastava is just one (beautiful and useful) manifestation of it. I can see why the practical algorithm would work even if it cannot be explained formally via spectral sparsification, and including the slow unimplemented algorithm just for the sake of its analysis feels a bit forced. What does it add to our understanding, and was it worth making the paper that much more confusing? Ultimately it's the author's choice, but even now the way the writeup deals with the two algorithms is still "evolving", and it is not clear how a final version would look. (I don't think the current form makes sense, since the algorithm now titled "FastGAT" is not faster than GAT, and section 3.1 zigzags between the two algorithms somewhat awkwardly.)

Nevertheless, in the end the authors were straightforward about all this in the discussion. As I said originally, I like the overall approach, so as long as the clarifications about the gap between the analysis and the implementation are included, and pending other reviewers' concerns about novelty and experimental validation (I am less versed in the empirical literature on GNNs so prefer to defer to them on those points), I think the paper could still be accepted.

---

> ### Author Response · Authors · 2020-11-22
> **Thank you for the careful reading and understanding! We clarify our algorithm below.**
>
> ** For comments on Section 3: **
>
> 1. A is the weighted adjacency matrix.
>
> 2.  (Clarification for questions 2. and 3. ) For theoretical purposes, we assume a layer-wise sparsification process: The attention weights computed across all the edges act as the new edge weigths, and the reweighted edges after sparsification form the new set of attention coefficients used to compute the feature updates. We state this layer-wise assumption in theorems. Hence, our theory holds true, as the output weights are used for computing the GAT updates.
>
> In practice however, this would not result in a gain in computational complexity. In our experiments, we use variations of the algorithm by either using a single sparsified graph for all layers and attention heads (FastGAT-1-const) or using a different sparsification in each layer (FastGAT-layer). For both of these variations, we use the input edge weights for sparsification.
>
>
> Further, we acknowledge that the theorems assume a layer-wise sparsification process, which is different compared to the experiments. However, this gap is due to the following problem, which to the best of our knowledge is still an open research problem: For a graph G, a set of edge weights W and a corresponding spectrally sparsified graph G', for what perturbations of W is G' still a valid spectral sparsification? This depends on the both the transformation applied to W and the graph G itself.
>
> ** Comments on section 4: **
>
> Although our theorems follow from previous results on spectral sparsification, they provide a justification of why using spectral sparsification is a good idea for convolution based GNNs, making our approach less heuristic.  Further, as we pointed out above, by studying perturbation properties of spectral sparsifiers, less restrictive theoretical analysis might be possible, but we leave this for future work.
>
> We appreciate the set of minor comments by the reviewer and we have updated our main submission file accordingly.
>
> We have updated our main file to reflect all of the above comments. We find this review very helpful and we thank the reviewer for a careful reading and understanding of our paper.

---

> > ### Comment · AnonReviewer2 · 2020-11-23
> > **Response**
> >
> > Thank you for the answers and revisions. I'm still not sure about where/how the weights of the sparsified graph come into the computation, I would appreciate if could clarify again (perhaps you could point to the exact place in the specification of Algorithm 1 where they are used). I think I'm confused since on one hand it looks like the attention coefficients are computed by the attention function, on the other hand you write that the attention coefficients are set to the sparsified graph weights, and I don't yet see how these fit together.
> >
> > Also I see now there is a hat over the $\Gamma_k^{(\ell)}$ in the last line of the inner loop, what is the relation between $\Gamma_k^{(\ell)}$ and $\hat\Gamma_k^{(\ell)}$? (Could not find it in the paper, apologies if I've missed it)

---

> > > ### Author Response · Authors · 2020-11-24
> > > **Clarification on the algorithm**
> > >
> > > We realized that our statement of the algorithm in the paper was causing some miscommunication. We have updated the algorithm algorithm statement in the paper and we provide the clarification below as well:
> > >
> > > The most general form of the algorithm, for which we present our theoretical analysis is the following:
> > >
> > > In each layer and attention head, compute the attention coefficients for all the edges, use the attention weights as input the sparsification algorithm, and use the re-weighted attention coefficients of only the remaining edges for the feature update. For this setting, the theoretical analysis holds.
> > >
> > > However, the above algorithm is clearly expensive and does not provide any enhancement in terms of speed. So in practice, we use the input edge weights for sparsification, and do not further consider the reweighting of the edges. We then only compute the attention coefficients for the remaining edges, which contributes directly to the gain in speed.
> > >
> > > We realize the gap between the general algorithm for which we provide theoretical analysis, and the practical algorithm we use. This gap arises due to the open problem we described in the previous comment. Further, the simple one-time sparsification still achieves good empirical performance, as shown in the paper.
> > >
> > > We hope this clarification helps. We are happy to answer any further queries. We again thank you for the careful reading of the paper.

---

### Official Review · AnonReviewer3 · 2020-10-25
**Limited novelty**

**Rating:** 4
**Confidence:** 5

**Review:**

This work proposed to use graph-sparcification method to decrease the number of edges so as to support GAT training over large networks. The graph sparcification approach is based on resistance. Empirical results by comparing with standard GAT or randomly edge-dropping strategy demonstrate the effectiveness of their approach.

This work is written very well. The logic flow is good. I also appreciate the experiments as the authors tend to demonstrate their method in a comprehensive way. However, I think that this work is only with limited novelty given some previous works and therefore cannot give a recommendation.

1. This work seems to simply combine two irrelevant techniques together to make it read novel. However, there two techniques seem to be contradictory in some sense: Obviously, the graph-sparsification method in this work is independent of the GAT building block. I believe that it can be also used to improve GCN, which also works well as long as the networks are homophilic as used in the evaluation sections. Moreover, as the graph sparsification capture the edges that emphasize more on network centralities, it will naturally reduce the advantage of GAT over GCN, when they are both combined with this graph sparsification. This is because the graph sparsification naturally performs some selection on effective edges just as GAT did. So in my mind, perhaps, graph sparsification + GCN have already done a very good job and have much less complexity, where attention is never needed. If we ignore the attention part, it is actually not novel to user graph diffusion (including resistance used in this work which is nothing but a more complex result of graph diffusion (commute time)) to revise graph topology and perform graph sparsification. Please check [1]. Given the above points, I did not see much novelty and rationality from this work.

2. The theoretical analysis is not novel, which is just some easy analysis regarding spectral graph theory. Moreover, GCN has a tighter bound than GAT, which may also be related to my concern in the first bullet. I think the only valid theoretical results are to demonstrate the end-to-end performance instead of just one-layer distortion.

[1] Diffusion helps graph learning. NeurIPS'19. (and its references.)

---post-discussion update---
I greatly appreciate the authors' effort to actively attend the discussion. In general, I like the idea to leverage the graph sparsification technique to accelerate GNN training.

However, I think there are still several fundamental questions that should be well addressed before this paper get accepted.  First, I do not see clear reasons to emphasize a specific model GAT. One concern arises during the discussion, where the sparsification technique here only works for GAT instead of GCN or other GNN smoothing models, which brings me some concerns about the technique. Second, the fundamental difference between this method and GDC [1] is still not clear. Both methods emphasize the low-frequent section of the graph connection. Why does the sparsification method here work while GDC does not work, though the sparsification method here can also be viewed as a type of graph diffusion (GDC)?

---

> ### Author Response · Authors · 2020-11-22
> **Thank you for the review and a note on the relevance of the two techniques**
>
> We wish to clarify the key insight behind our paper: We identify the connection between convolution based GNNs and spectral sparsification. In fact, there are many graph sparsification methods that aim to preserve a ``suitable distance metric" between the original and sparsified graphs. Our choice of preserving the spectral properties are specifically motivated by the convolution operations that form the backbone of attention based GNNs. In this regard, combining the two techniques was a deliberate choice, as otherwise, any existing sparsification technique could have been used.  In a more general sense, the techniques have more to borrow from each other. For example GNNs can motivate theoretical studies. In particular, our experiments have helped us identify the following open problem: For a graph G, a set of edge weights W and a corresponding spectrally sparsified graph G', for what perturbations of W is G' still a valid spectral sparsification? Conversly, existing theory on graphs can inform the practice of GNN models, as shown in this paper. In fact, the experimental results in our paper clearly show that the two sets of techniques are not irrelevant to each other and that our proposed method could be an effective tool for practitioners.
>
>
> **With regards to sparsification reducing the advantage of GATs over GCNs:** We have clear experimental evidence that sparsification does not reduce the advantage of GATs over GCN style models.109FastGAT ourperforms GCN style models, as shown in the paper. Although the argument presented by the reviewer110seems to be well stated, our extensive set of experiments presented in the paper prove otherwise.
>
> ** Comparison to "Diffusion improves graph learning" ([1]):**
>
> We thank the reviewer for pointing out this paper, we enjoyed reading it and the ideas presented are indeed related. However, we would like to share the differences between out paper and [1]:
>
> 1. GDC ([1]) does not result in a much sparser graph. The sparsification is applied to the diffusion matrix, which typically is dense, as pointed out in [1] itself (Section 3, [1]). Further, Figure 5. in [1] shows the average node degrees of the diffusion matrix **after sparsification** in their model that is required to achieve good performance on 3 datasets. Note that this average degree is around 40, which is actually higher than the underlying node degree (B.1, [1]). To summarize, **[1] increases the graph desntiy via diffusion and then sparsifies it, to address classification performance**, whereas **our paper goes in the opposite direction by just sparsifying the original graph, to address computational complexity**. The goals of the two papers, and the underlying methods are in fact different.
>
> 2. We are able to apply GAT models to **large scale datasets**. Note that [1] does not provide experimental results on datasets such as Reddit, which is at a much larger scale.
>
> 3. In [1], the authors note that their method does not directly preserve the spectrum of the graph (Page 5, [1]). Our method is explicitly meant for preserving the spectrum.
>
> In general, we find that the goals of the two papers are very different: this is especially clarified by the fact that we129provide experiments comparing training times whereas they do not
>
> ** With regards to theoretical analysis: **  In general, we formulate the problem of using spectral graph sparsification in the context of GNNs. We acknowledge that out theory is not ``end-to-end". However, the theorems presented in the paper lend support to our intuition and to the strong experimental results. In this sense, our method is backed by theoretical insight, rather than just being a heuristic method.
>
> Overall, Our method can be directly used by practioners as a tool to improve the efficiency of training GNN models. It is simple enough to be augmented quickly to GNN models, while enjoying strong performance. It also opens up a connection between theoretical constructs such as spectral sparsification and ML practice. In this sense, we believe that our method is useful and novel.

---

> > ### Comment · AnonReviewer3 · 2020-11-23
> > **Further comments**
> >
> > Many thanks for the detailed response from the authors. However, I still have the following criticisms and questions.
> >
> > 1. I did not argue that the combination of "graph sparsification" and "GNNs" is not useful. The issue is that this idea is not novel, as previous work GDC has already introduced. Effective resistance based graph sparsification is also a graph-diffusion sparsification, as it simply works on specifying the pseudo inverse of normalized Laplacian matrix, which is just graph diffusion "I + \hat{A} + \hat{A}^2 + ....", where \hat{A} = D^{-0.5}AD^{-0.5}. Therefore, the idea is not novel.
> >
> > Thanks to the authors' clarification, I understand that this work essentially used a weighted and sparse version of the original graph structure. But this is not enough significant given that the idea is still from graph diffusion.
> >
> > 2. The comparison between GCN and GAT is unfair. FastGCN [Chen et al. 2018], cited by the authors is not what I asked for. I thought the gain by simply replacing GAT with GCN in the graph diffusion based sparsification framework (like GDC or this work) would diminish. This is where the authors would like to make some comparison.
> >
> > 3. Regarding the practical aspect of the model, I am still worried. This method proposed here emphasizes too much on the low frequent side of the graphs, which is only applicable to predict homophilic node labels, which essentially limits the usage of GNNs.

---

> > > ### Author Response · Authors · 2020-11-24
> > > **Further clarification**
> > >
> > > Dear reviewer,
> > >
> > > We thank you again for the discussion, we do believe that comparison to GDC is a relevant point of discussion. We clarify a few pointes below:
> > >
> > > 1. The fundamental difference between out paper and GDC is the ** purpose**. Please note that the GDC paper does not make any claims on improving the training/ testing complexity. In fact, as we pointed out in our previous comment, the average degree of the nodes used in the GDC paper is ** higher ** than the original average degree, which clearly ** increases** the training/testing time.  On the contrary, our primary goal is to achieve a speed-up. We understand that the underlying ideas are comparable. But please note that in GDC, diffusion is used as a *tool to combine spatial and spectra methods to achieve better classification performance*. In fact, GDC is not a standard baseline in GNN related papers that study the computational complexity. We did not derive our idea from GDC and we do not see our paper as a specific instance of GDC.
> > >
> > > 2. Thank you for clarifying your query. We in fact performed experiments by replacing GAT with GCN and then using graph sparsification. We found that the performance of the sparsification+GCN model equals or slightly underperforms the original GCN model. We also tried sparsification+FastGCN. In this case also, the performance was equal or slightly under that of the original FastGCN itself. We hence present comparison with respect to only FastGCN as it is also a standard baseline. In other words, even though the intuition behind the argument presented by the reviewer seems plausible, the experimental results simply contradict it. We hope we have convinced the reviewer that sparsification+GAT ** does not** result in a diminishing of its advantage over GCN.
> > >
> > > 3. We would like to address this comment by simply stating that our proposed method works well on an extensive set of real world graphs, as shown in the paper. The degree to which these graphs are homophilic is something we do not study in our paper, similar to most papers in the GNN related literature. We only claim to perform as well as state-of-the-art GNN papers in terms of classification performance on the common set of datasets found in the relevant literature.
> > >
> > > Thank you again for the interesting discussion.

---

> > > > ### Comment · AnonReviewer3 · 2020-11-24
> > > > **Further comments**
> > > >
> > > > Many thanks for the authors' actively participating into the discussion.
> > > >
> > > > 1. Regarding the novelty as opposed to GDC, I do not think the difference in the purpose is a "fundamental" difference. Both ideas follow the logic that the low-frequent spectral components of the graph may help with the node classification. GDC perhaps views the graph sparsification as a side point while this work views it as a more core contribution. This is just related to different emphasis instead of techniques. For the next version of this work, I suggest authors conduct a good study on why the sparsification adopted in this work is better than that adopted in GDC, which would help with establishing the contribution of this work.
> > > >
> > > > 2. For the experiments on GAT vs GCN, I still believe in theory and think the empirical results depend more on the effort of parameter tuning. Moreover, if the authors indeed observed that issue, does it mean that the sparsification unfortunately limits the usage of different GNNs, say other models, GraphSAGE, GIN, etc.? I suggest that the authors should look into this issue in the next version of this work. Sorry for being tough. I prefer to make the direction of GNN research in a  more rigorous way. Before claiming something, I would like to first understand it.
> > > >
> > > > 3. Regarding the datasets, I see it again an issue of current GNN research. There are some non-homophilic datasets in GeomGCN [1] and distance encoding [2]. But perhaps, I have asked so much for this work to also perform well on these datasets.
> > > >
> > > > Overall, if the authors may conduct a good study of points 1,2, I would like to support this work next time.
> > > >
> > > > [1] Geom-GCN: Geometric Graph Convolutional Networks
> > > > [2] Distance Encoding: Design Provably More Powerful Neural Networks for Graph Representation Learning

---

> > > > > ### Author Response · Authors · 2020-11-24
> > > > > **Further clarification**
> > > > >
> > > > > Dear reviewer,
> > > > >
> > > > > Thank you again for the discussion.
> > > > >
> > > > > 1. Regarding "why sparsification adopted in this work is better than GDC": Please note that we do not make any claims about our method being better in terms of the classification performance. In this regard, our claim is only that we do equally well. However, our claim is that we do much better in terms of the computational complexity of training/ inference. The reason for this is straight-forward: The average node degree after sparsification in our method is much lesser than the original average degree. In GDC, note that the authors use a higher order of diffursion (K). This renders the graph denser than the original graph. This is why we feel the aims of the paper and the way we use sparsification is much different. As for "GDC perhaps views the graph sparsification as a side point", please note that GDC does not study the time aspect at all. We study just that. So the two papers are orthogonal in this sense.
> > > > >
> > > > > 2. Our main claim is the following: Given a convolution based GNN model, sparsification will retain the accuracy of the model, while being faster. The underlying model maybe GraphSAGE, GCN, GAT, etc. So sparsification does not limit the usage of a model, as long as the same model is the choice prior to sparsification. We mainly study sparsification in the context of GATs owing to the high computational complexity associated with it. It is this computational bottleneck of the GAT model that drives our choices, rather than any limitation of the sparsification process.
> > > > >
> > > > > 3. We do think that non-homophilic datasets are very interesting to study and we thank you for pointing us in this direction. However, as you noted, it's not the focus of our paper, similar to existing GNN literature. In this regard, we do not consider the lack of study on non-homophilic datasets as a drawback of our paper, similar to existing GNN literature.
> > > > >
> > > > > Thank you.

---

### Official Review · AnonReviewer1 · 2020-10-28
**Impressive results on scaling up GATs**

**Rating:** 6
**Confidence:** 3

**Review:**

The authors propose FastGAT, a methodology for guided sparsification of graphs such that they will largely preserve performance of graph attention networks (GATs), while vastly improving on their computational performance.

The sparsification is based on the spectral properties of the input graphs, and seems to be backed by strong theorems that dictate the upper bound on the distance between the computed features and full-graph features. Results on many datasets seem to strongly back this, and demonstrate that such a sparsifier indeed performs better than purely randomised sparsifying.

It looks like an interesting and novel paper that could enable large-scale applications of attentional GNNs. I like that authors surveyed several attentional mechanisms (e.g. AGNN and GaAN) and confirmed their findings across them. I recommend acceptance, and would invite the authors to consider the following:

- The authors mention that GraphSAGE is not amenable to attentional models, but in principle, isn't GraphSAGE style update (node batching + subsampling neighbours with replacement) what GaANs already usefully applied to Reddit? If there are no clear reasons why GraphSAGE sampling + GAT is not applicable, maybe it should be included as a baseline.
- Typos: "Eq.equation 1", "We compar"

============= Post-rebuttal:

I thank the authors for their careful responses. After discussing with other reviewers:
- I agree that the sparsification method proposed here is also in principle applicable to GCN-like models;
- The authors should have provided results for "FastGAT"-style sparsification on GCN, rather than countering the reviewers using passages like "Hence, it is mainly the question of necessity rather than applicability which guided our choice of studying the GAT model in depth."
- If such a GCN model ends up competitive, the focus of the paper could switch to the sparsification method itself rather than the GNN model it is applied to.

In light of these discussions, I am decreasing my score to a weak accept, and I hope the authors will take this advice for the next iteration of their work (which otherwise, in my opinion, deserves being published in a strong venue).

---

> ### Author Response · Authors · 2020-11-22
> **Thank you for the kind comments**
>
> We would like to clarify this statement: We only point out that GraphSAGE method directly does not use the attention mechanism by itself. Extention of such GraphSAGE style sampling and aggregating has indeed been studied in the GaAN paper.
>
> We thank the reviewer for the careful reading of the paper and we have fixed the typos.

---

### Official Review · AnonReviewer4 · 2020-11-05
**Good empirical results, more discussions needed on sparsification family and transferrableness of this approach**

**Rating:** 5
**Confidence:** 4

**Review:**

## Summary
This paper proposes a paradigm which speeds up the training/inference time of GATs while not compromising too much performance. The method adopts a layerwise sampling procedure. In particular. The authors propose to sample a sub-portion of edges for each layer based on their effective resistance. Such sampling keeps the spectral similar to the original results theoretically and gives a guarantee to the performance drop.

## Reasons for Score
The paper discusses the important topic of fast training and inference and proposes a sampling strategy over edges that have not been discussed (e.g. previously mostly on nodes). My main concern is about the novelty (a direct application of sparsification) without too much discussion about the broader sparsification family and the whether this technique is universal enough on other GNN frameworks.

## Pros
The paper tackles the problem of fast GAT training/inference, which is an important problem when training large-scale networks. The proposed method shows good empirical results with a proper theoretical guarantee.

## Cons
1. Novelty. The overall framework seems to be a direct application of effective resistance-based edge sampling. The theorem followed-up seems a direct result which makes the novelty limited. With this being said, a more meaningful investigation (both in the related works or simple experiment justification) of the broader area of (weighted) spectral sparsifier may be discussed.

2. Applicable to other GNN networks. While the resistance-sampling strategy seems a standalone strategy and not particularly bound to GAT, whether this paradigm can be a universal technique in other GNN networks (e.g. plain GCN) is worth investigating.

3. The comparison of FastGCN [1] w.r.t performance may not be that fair. The barebone framework of FastGCN and this work FastGAT differ, it might be meaningful to compare the metrics under the same setting (both GCN or GAT). Also, some other node-sampling procedures might be meaningful baselines (e.g. Cluster-GCN [2]).

[1] FastGCN: Fast Learning with Graph Convolutional Networks via Importance Sampling
[2] Cluster-GCN: An Efficient Algorithm for Training Deep and Large Graph Convolutional Networks

## Post rebuttal
Thanks for providing detailed explanations.

Overall, I think the authors tried hard to prove the concept of “edge” sparsification helps for speedup of “attention” GNNs which I also think they explanations/rebuttals succeeded in doing this, though the established theory seems quite standard and is not the same as in the experiments.

However, pertaining the results I still do not see a claimed comparison of GCN, “FastGAT”-sparsified GCN in the revised work based on the replies to R3 (only FastGCN is reported. Note this is not a direct adaption of the authors’ method, but from the previous node-sampling literature). As is claimed by the authors, performing sparsification on GCN does not provide seminal speed boost. However, while GCN is a strong/simple baseline without heavy parameter tuning, it’s hard to make a clear justification on why a sparsified heavy-attention-computation network (FastGAT) would do any good if its final results are barely comparable/similar to a naive GCN baseline with a similar running time.

Given the idea of sparsification is not novel which I stand on a similar point with R3 and specifically with its linkage to GDC. I think the current paper may need to be improved with some more evidence on proving the “edge” sparsification method is superior to other “node” sparsification versions of attention networks e.g. empirically/theoretically better (“FastGCN”-sparsified, “ClusterGCN”-sparsified models) on a complete set of datasets; or with a more rigorous comparison to other simple GCN-versioned sparse methods that do not need the heavy attention computation in the first place.

For the current status, I would still lean towards a rejection.

---

> ### Author Response · Authors · 2020-11-22
> **Thank you for the review**
>
> **Novelty:**
> The novelty of our paper lies in the following: Our goal is to not just show that using effective resistance sampling can speed up GAT models. Our main insight is that since graph networks are based on graph convolution, spectral sparsifiers in general might be useful tools in accelerating training/ inference. We demonstrate this idea by using the specific spectral sparsifier proposed by Spielman et. al. applied to the computationally expensive graph attention networks. Admittedly a simple method, the experimental results show that it can be a very useful to practitioners.
>
> Note that we provide a summary of other graph sparsification techniques in the related work section (Section 2) and we also clarify why we choose spectral sparsification over others. Further, note that the ides of using cut-sparsifiers (that preserve weights of the edges along any ``cut" of the graph) is a special case of spectral sparsification. In general, spectral sparsification has the strongest guarantees in preserving graph structure, and their use in GNNs is well motivated because of the convolution operations that directly depend on the eigenvalues of the graph. This is our motivation to choose the technique of spectral sparsification.  If the reviewer would kindly point us to the specific weighted spectral sparifiers to be discussed, we would be happy to include that.
>
> **Applicability to other GNNs: **
>  We appreciate this suggestion and it was one of our own considerations. However, note that GNN frameworks not based on attention, including GCN, are already computationally efficient. In fact, GCN updates can be done using fast matrix multiplication libraries. Although spectral sparsification will result in good learning performance, it may not add value in terms of computational complexity. In this regard, our choice to not include a study on GCNs is simply because there not much computational advantage. On the other hand, we choose to study the GAT model due to the high computational burden of computing pairwise attention coefficients in every layer of the network and forward pass of the algorithm. Hence, it is mainly the question of necessity rather than applicability which guided our choice of studying the GAT model in depth.
>
> **  Comparison to GCN: **
> We included GCN and FastGCN as baselines as they are standard baseline methods for semi-supervised node classification tasks. Further, we also wanted to demonstrate that graph sparsification does not negate the advantage of attention over simpler GCN style models.
>
> The main goal of our paper is to show that expensive GNN models such as the GAT model can be sped up using graph sparsification. Having said that, spectral sparsification is also directly applicable to other GCN based models such as FastGCN and Cluster-GCN, as they are also convolution based GNNs. In general, given a convolution based GNN framework, spectral sparsification can be used as  tool to make the model lighter, while preserving the learning performance.

---

### Public Comment · ~Benedek_Rozemberczki1 · 2020-11-10
**Erroneous attribution of the GitHub dataset**

The Github dataset was not created by the authors of DGL. The appropriate citation is:

>@misc{rozemberczki2019multiscale,
       title = {{Multi-scale Attributed Node Embedding}},
       author = {Benedek Rozemberczki and Carl Allen and Rik Sarkar},
       year = {2019},
       eprint = {1909.13021},
       archivePrefix = {arXiv},
       primaryClass = {cs.LG}
       }

---

> ### Author Response · Authors · 2020-11-22
> **Updated citation**
>
> Thank you for the clarification, we have corrected the citation in the updated manuscript.

---

### Decision · Program_Chairs · 2021-01-07
**Final Decision**

**Decision:**

Reject

**Comment:**

Two reviewers recommend rejection, whereas two reviewers slightly lean towards acceptance. All reviewers agree that the paper tackles an important problem, and the proposed direction holds promise and is worth exploring. However, the reviewers raised concerns about the novelty of the proposed approach [R3,R4], the applicability of sparsification to GCN-based models [R3,R4], baseline experiments [R1,R3,R4] and the gap between the theoretical aspect of the paper and the implementation of the proposed approach [R2]. The authors engaged with the reviewers during the discussion period and succeeded in motivating the speedup gains of their method, and clarifying some of the reviewer's concerns. However, after discussion, the reviewers still think this is a borderline paper, which could be significantly strengthened by validating the applicability of the proposed sparsification to other GNNs [R1,R2,R3], and in particular, by including the suggested FastGAT-sparsified GCN experiment [R1,R3,R4]. The paper could also benefit from improving the presentation of both the analyzed approach and the practical one [R2]. I agree with reviewers' assessment and therefore must reject. However, I acknowledge that the paper does raise notable interest and I encourage the authors to consider the reviewers' suggestions in future iterations of their work.